# OptiMUS: Optimization Modeling Using MIP Solvers and Large Language Models

## Abstract

Optimization problems are pervasive across various sectors, from manufacturing and distribution to healthcare. However, most such problems are still solved heuristically by hand rather than optimally by state-of-the-art solvers, as the expertise required to formulate and solve these problems limits the widespread adoption of optimization tools and techniques. We introduce OptiMUS, a Large Language Model (LLM)-based agent designed to formulate and solve MILP problems from their natural language descriptions. OptiMUS is capable of developing mathematical models, writing and debugging solver code, developing tests, and checking the validity of generated solutions. To benchmark our agent, we present NLP4LP, a novel dataset of linear programming (LP) and mixed integer linear programming (MILP) problems. Our experiments demonstrate that OptiMUS solves nearly twice as many problems as a basic LLM prompting strategy. OptiMUS code and NLP4LP dataset are available at https://anonymous.4open.science/r/nlp4lp-8F62/README.md

## 1 Introduction

Optimization problems are common in many fields like operations, economics, engineering, and computer science. Important applications of optimization include reducing energy use in smart grids, improving supply chains, or increasing profits in algorithmic trading (Singh, 2012; Antoniou & Lu, 2007). Major advances in optimization algorithms over the last several decades have led to reliable and efficient optimization methods for a wide variety of structured optimization problems, including linear programming (LP) and mixed-integer linear programming (MILP) among many others. Unfortunately, optimization modeling — transforming a business problem into a mathematical optimization problem in standard form — still requires expert knowledge. This expertise gap prevents many organizations from using optimization, even when it could significantly improve their operations. E xamples include inventory management in supermarkets, patient operations in hospitals, transportation policies in small municipalities, energy management in local solar farms, and operations in small businesses or NGOs (Saghafian et al., 2015; Aastrup & Kotzab, 2010; Yao et al., 2020; Shakoor et al., 2016). Automating optimization modeling would allow sectors that can not afford access to optimization experts to improve efficiency using optimization techniques.

Large language models (LLMs) offer a promising way to make optimization more accessible. LLMs have demonstrated the capability to understande, generate, and interpret natural language for many tasks. They make it easier to formulate problems and set up solutions, making expert knowledge more widely available. However, the role of LLMs in the optimization landscape is still unexplored, mainly due to their novelty and the absence of comprehensive benchmarks. To explore the capabilities and limitations of LLMs in optimization, this paper makes the following contributions:

- We introduce a novel dataset, NLP4LP, comprising human-expert formulations of 52 LP and MILP optimization problems, annotated with their solutions, code to check optimality, and sample formulations of the problem in markdown and in code. To construct this dataset, we introduce a standardized format to represent optimization problems in natural language.
- We present OptiMUS, an LLM-based agent to formulate and solve optimization problems. Fig. 1 demonstrates the structure of OptiMUS.

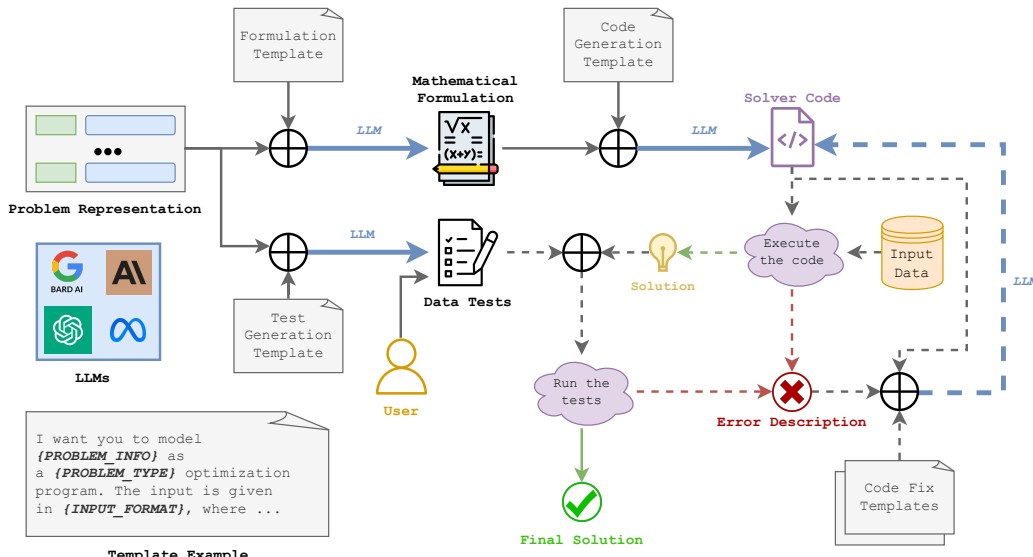

Figure 1: An illustration explaining how OptiMUS uses various components to effectively model and solve optimization problems. First, a mathematical formulation is generated from the problem representation. The solver code is then generated based on the formulation. The code is executed to generate and save a solution to file. The solution is then tested on a set of unit tests generated by the LLM and revised by the user. If the code does not run or fails the tests it is passed to the LLMs along for the relevant error code for revision until it is fixed (dashed lines might be executed multiple times). Otherwise, the output is selected as the final solution. An example template is shown in the bottom left corner.

- We develop techniques to improve the quality of OptiMUS and demonstrate their effectiveness, including automated data augmentation via problem rephrasing and self-fixing of solutions via automated testing and debugging. Using these techniques, OptiMUS increases the solve rate by 91% compared to direct prompting.

By integrating the capabilities of LLMs with optimization techniques, our work aims to democratize access to optimization across application domains, extending the reach and utility of optimization.

This paper builds on recent progress in Large Language Models (LLMs) and optimization. A more comprehensive review of this work is deferred to Section 6, and we herein on ideas most closely related to our topic. In a very recent paper, Chen et al. (2023) develop a chatbot to help users detect and fix infeasible optimization problems. In contrast to our work, their agent takes a `Pyomo` code rather than a natural language description as input, and acts as an AI assistant rather than as a solver. (Yang et al., 2023) use LLMs to directly generate solutions to optimization problems, focusing on the problem of modifying prompts to improve performance. In contrast to our work, their method does not rely on solvers, coding, or any other tools. Their model can not address problems with medium or large input data sizes because 1) even the context size of LLMs is very limited compared to the input data size of many optimization problems and 2) LLMs' performance substantially decreases as the input context grows longer, even for explicitly long-context models (Liu et al., 2023).

This paper is organized as follows: Section 2 discusses the challenges of using LLMs to solve optimization problems; Section 3 describes the details of our LLM-based optimization agent; Section 4 outlines the dataset creation process and statistics; Section 5 presents the experiments and analysis; Section 6 explores the related work; and Section 7 concludes the paper with future directions and implications. The appendix includes prompts, details on the experiments' setup, and further analysis.

## 2 Challenges of Optimization Modeling using LLMs

Optimization problems are defined mathematically by an objective function and a set of constraints. For example, an MILP can be written as

$$
\begin{aligned}
\text{minimize} \quad & \boldsymbol{c}^T \boldsymbol{x} \\
\text{subject to} \quad & \boldsymbol{A}\boldsymbol{x} \leq \boldsymbol{b} \\
& x_i \in \mathbb{Z}, \ i \in \mathcal{I}
\end{aligned}
$$

An optimization workflow consists of 1) formulating an optimization problem in standard form by identifying its objective and constraints, and then 2) solving the problem, generally using code that calls an optimization solver. Formulation is often a challenging task even for experts in optimization. Different formulations can lead to significantly different solving times and enable the use of different solvers or solution techniques (Boyd & Vandenberghe, 2004). One important skill for an optimization expert is to identify assumptions or relaxations that allow for casting the problem as a well-studied problem type, such as MILP, which enables the use of well-developed optimization solvers. Crafting such an efficient formulation often requires specialized knowledge (Zohrizadeh et al., 2020; Low, 2013; Roubíček, 2020; Luo et al., 2010; Jakob & Pruzan, 1983).

Given the formulation, an optimization expert must choose a solver. Each solver has its own interface and functionalities (Achterberg, 2019; Diamond & Boyd, 2016; CPLEX User's Manual, 1987). However, user manuals for these solvers are often hundreds of pages, making them hard to fully understand. This complexity often makes it challenging to choose the right solver and arrive at a high-quality implementation, much less to switch solvers in order to find the one best suited for a given problem.

Recent progress in Natural Language Processing (NLP) has led to the development of large language models (LLMs) useful in various tasks like answering questions, summarizing text, translating languages, and coding (OpenAI, 2023; Touvron et al., 2023; Chowdhery et al., 2022; Wei et al., 2023). Connections to other software tools extends the reach and accuracy of LLMs, as demonstrated by plugins for code writing and execution (Paranjape et al., 2023; Wei et al., 2023). Consequently, leveraging LLMs for optimization modeling is a logical progression. A basic approach is to describe a problem to an LLM and prompt it to write the solver code. However, this direction faces several key challenges:

- **Variable Representations:** An optimization problem can be described in many ways. For example, a user might use different terms (vehicle vs. car vs. truck vs. carrier), notations (*price* and *capacity* vs. $p$ and $c$ vs. $x$ and $y$), or even omit assumptions based on common sense (capacity of a vehicle is nonnegative, number of employees is an integer, etc.). While humans can easily adapt to these variations, LLMs may struggle depending on their training data and methods.

- **Handling Large-Scale Problems:** LLMs have a limited context size. Even for long-context models, performance substantially decreases as the input context grows longer (Liu et al., 2023). However, the specification of an optimization problem often involves a large amount of numerical data, such as attributes of customers or of goods sold, which could easily require gigabytes of memory.

- **Unreliable Outputs:** The solutions provided by LLMs are not always reliable. The generated code may be incorrect or even not executable. It is especially challenging to verify the solution when the code runs but the output is incorrect. For instance, if the code runs and claims the problem is unbounded, perhaps a constraint has been accidentally omitted from the formulation? Moreover, writing tests to check the correctness of the output can also be complicated and time-consuming.

Solving optimization problems with language models requires a multi-faceted approach combining advances in natural language processing, optimization theory, and software engineering. The following sections introduce our LLM-based optimization agent OptiMUS, demonstrate how OptiMUS mitigates these challenges, and present empirical evidence of its effectiveness and limitations.

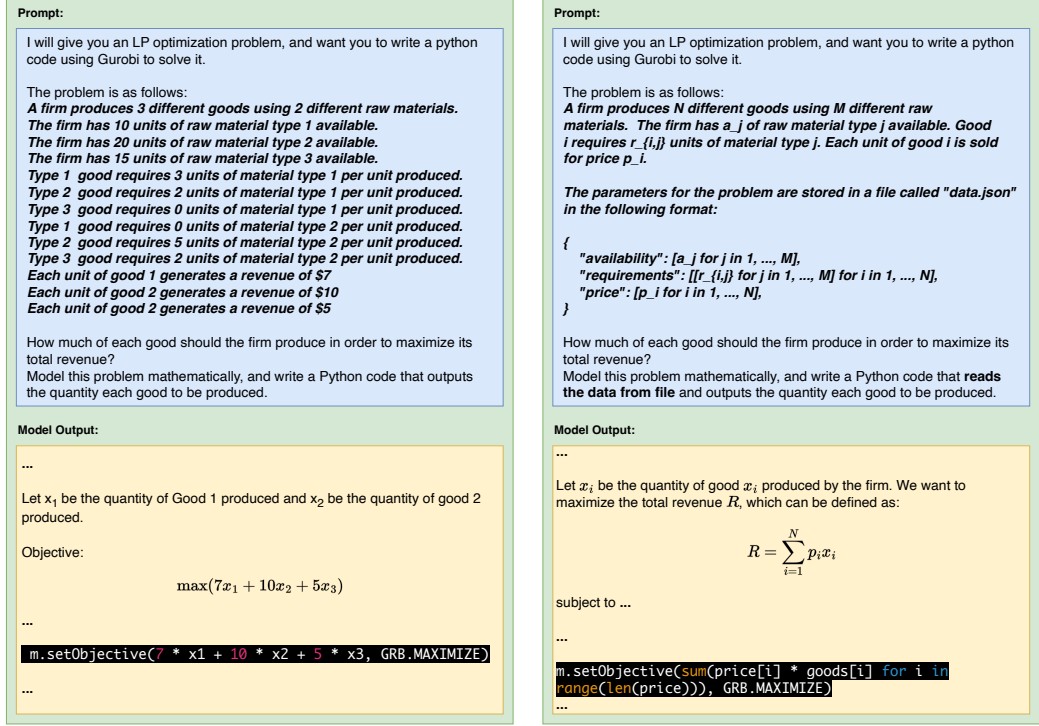

Figure 2: Scaling OptiMUS to problems with large numerical data: Instead of passing everything to the LLM directly (left), in OptiMUS we separate the numerical data from the problem description and give the metadata to the LLM (right). The LLM then writes a code to interact with the data file.

## 3 METHODOLOGY

This section details the design of OptiMUS. See Fig. 1 for an illustration. OptiMUS starts with a structured description of the optimization problem, explained in Section 3.1, and a separate data file. It first transforms the structured description into 1) a mathematical formulation of the problem and 2) tests that check the validity of a purported solution. Afterwards, OptiMUS transforms the mathematical formulation into solver code. It joins the solver code with the problem data to solve the problem. If the code raises an error or fails a test, OptiMUS revises the code and repeats until the problem is solved or maximum iterations are reached. All prompts used by OptiMUS appear in Appendix A.

### 3.1 STRUCTURED NATURAL LANGUAGE OPTIMIZATION PROBLEM (SNOP)

As mentioned in 2, passing all the problem information directly to an LLM is not a scalable solution. To address this issue, we separate the data from the problem description, save the data as a `JSON` file, and then pass the format of this file to the LLM (an example is illustrated in Fig. 2). We use a standardized structure to organize our dataset instances, which we call Structured Natural language Optimization Problem (SNOP). OptiMUS takes a SNOP as input, and our benchmark dataset contains SNOPs and their solutions.

A SNOP has 6 fields (see Fig. 3):

- **Problem type**: The type of the problem (as a string), e.g., LP, MILP, QP, etc. For example, by targeting LP rather than MILP, a user can instruct OptiMUS to relax an integer problem. This field can be set to *ANY* to let OptiMUS decide how to formulate the problem.
- **Problem info**: A list of statements detailing the problem. We use $\backslash param\{\}$ symbol to identify problem parameters in the description. The optimization solver will replace these by the problem data after the problem is formulated by the LLM.

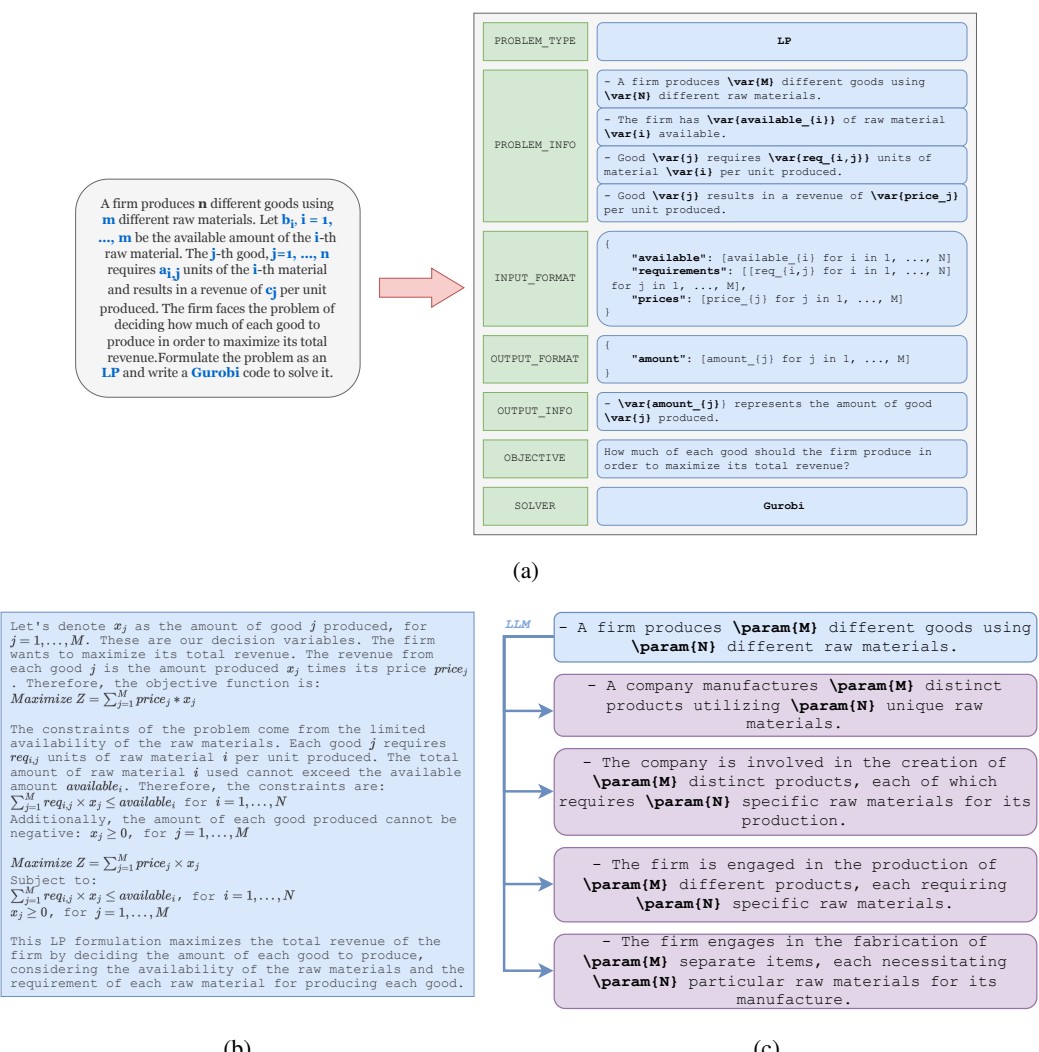

Figure 3: a). An example of a real-world optimization problem and a SNOP representation for it. b). An example markdown formulation of a problem generated by OptiMUS. c) Example rephrasings generated by OptiMUS from a problem info statement in the augmentation process.

- **Input format**: A string describing the format of the input data. We use `[]` to represent lists of values, names in quotes (`"`) to represent JSON keys, and `pseudo-for` to show indices.

- **Output info**: A list of statements describing the desired output.

- **Output format**: A string describing the format of the desired output.

- **Objective**: A string stating the objective of the optimization problem.

- **Solver**: The solver of choice (as a string), e.g., `Gurobi`, `cvxpy`, etc. This field can be set to *ANY* to let OptiMUS decide which solver to use.

Each problem in the benchmark has input data saved as a JSON file matching the input format. OptiMUS uses only the SNOP to develop a formulation, and then generates code to call an optimization solver, joining the formulation with the (possibly large) data to solve the problem.

```
...
for i in range(len(available)):
    constraints.append(cp.sum(requirements[i][j] * x[j] for j in range(M)) <= available[i])
...
```

```
...
for i in range(N):
    constraints.append(cp.sum([requirements[i][j] * x[j] for j in range(M)]) <= available[i])
...
```

Figure 4: OptiMUS prompts include instructions to avoid common coding mistakes. For example, ChatGPT commonly uses `cvxpy.sum` on generator objects instead of lists. Adding the instruction "- *cvxpy.sum takes a list as input, and not a generator*" to the code generation template reduces the incidence of this mistake. Top) generated code before the instruction; Bottom) generated code after adding the instruction.

## 3.2 FORMULATION

Given a SNOP, OptiMUS generates a mathematical formulation for the problem in markdown. At this stage, OptiMUS prompts the language model to define the optimization variables, the constraints, and the objective function.

## 3.3 CODE GENERATION

Given the problem formulation, OptiMUS generates `Python` code to read the input data from a `JSON` file, calls an optimization solver to solve the problem, and saves the output to another `JSON` file. OptiMUS uses `Gurobi` and `cvxpy` to solve the problems in our benchmark, and is capable of using advanced solver features. For example we observe it using `gurobi.abs_` to model $\ell_1$-norm objective instead of adding auxiliary constraints and variables. We observe that certain models make recurring mistakes when writing codes for certain solvers, and therefore our prompt includes solver-specific instructions to improve performance. For example, in `cvpxy`, the model commonly uses `cvxpy.sum` with generator objects instead of lists (see Fig. 4).

## 3.4 TESTS AND REVISION

Once solver code is generated, OptiMUS executes it with a `Python` interpreter. Two outcomes are possible: 1) an execution error occurs or 2) an output is written to file. Given an execution error, the error string is passed to OptiMUS along with the code and formulation, so that OptiMUS can revise the solver code to fix the error. This process is repeated until the code successfully executes or maximum iterations are reached.

Given an output, we need to ensure its correctness. OptiMUS generates unit tests using the SNOP to ensure the validity of the output. These tests check for 1) correct json formatting (e.g. the output json should contain "amount" as a key) 2) constraint satisfaction (e.g. detecting negative order quantities), and 3) consistency between the output values (e.g. sum of monthly profits should be equal to total profit). We cal these auto-generated tests.

See Fig. 5 for an example. Optionally, a user of OptiMUS can also revise auto-generated tests or write additional tests. We call these supervised tests. Our benchmark includes supervised tests for every problem. In our experience, developing supervised tests is roughly five times faster than developing equivalent tests from scratch. Given an output, OptiMUS runs the unit tests. Any tests that fail will generate error messages. OptiMUS uses these error messages to revise the code and fix it.

## 3.5 AUGMENTATION

As an additional strategy, OptiMUS automatically rephrases problems and attempts to solve the rephrased version using the same workflow above. If any of the rephrased versions succeeds, OptiMUS is able to use the solution and hence solve the problem. See Fig. 3c for an example.

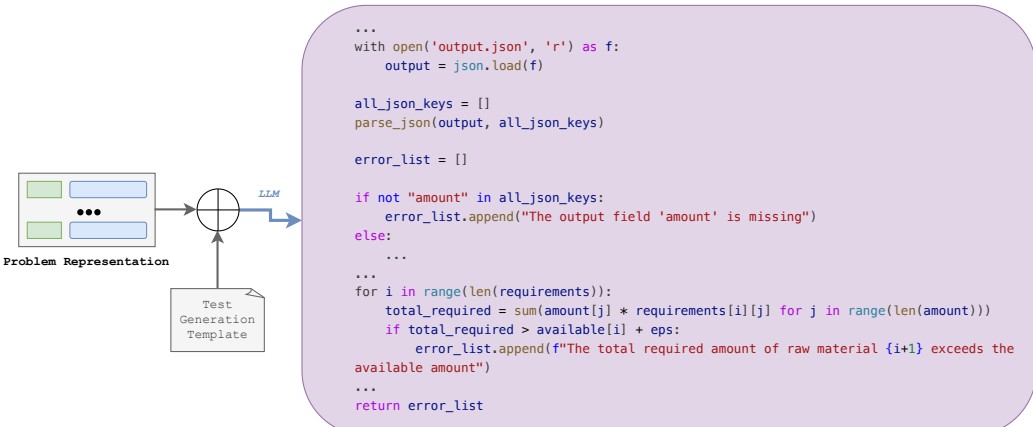

Figure 5: LLM can be used to generate tests and check the correctness of the output. After feeding the problem to an LLM using the test generation template, the model generates a script that checks the correctness of output (constraint satisfaction, output format, etc.) and returns appropriate error message if it finds a problem. The error message is used to automatically fix the code.

## 4 DATASET

To evaluate the performance of language models for solving optimization problems represented in natural language, we create a natural language optimization benchmark NLP4LP (Natural Language Processing for Linear Programming), consisting of 41 LP and 11 MILP problems (52 instances in total). The problems are drawn from textbooks and lecture notes in optimization (Bertsimas & Tsitsiklis, 1997; Williams, 2013; Nace, 2020). These resources appeared before 2021, and there is a chance that parts of these books have been discussed on the internet and used to train LLMs. However, none of these textbooks include code snippets. The natural language representations used in NLP4LP are further modified from the original problem statement by representation as SNOPs and by abstraction, as we replace numerical values in the original text by parameters. Moreover, it is clear from our results that the LLMs still find it challenging to formulate and solve these problems. The data consists of several views of each problem:

- SNOP (string) (see Section 3.1)
- supervised tests (code) (see Section 3.4)
- example data file (JSON) (see Fig. 8)
- optimal value (floating point number)
- sample optimal output (JSON) (see Fig. 8)

For each instance, we get the optimal value either from the textbook solution manual, or by solving the instance by hand. We have also included a test script for each instance for benchmarking purposes.

## 5 EXPERIMENTS AND ANALYSIS

In this section, we empirically evaluate OptMus on NLP4LP and identify its strengths and weaknesses.

We use GPT-3.5 and GPT-4 models for our experiments and adopt Gurobi as our optimization solver. Moreover, we allow a maximum of 5 iterations of revising the code. The task of developing optimization formulations and solver code directly from natural language representations is new and there are no baselines in the literature. Therefore, we use simple prompting as a baseline and analyze the effect of adding each component. Concretely, we consider these five modes:

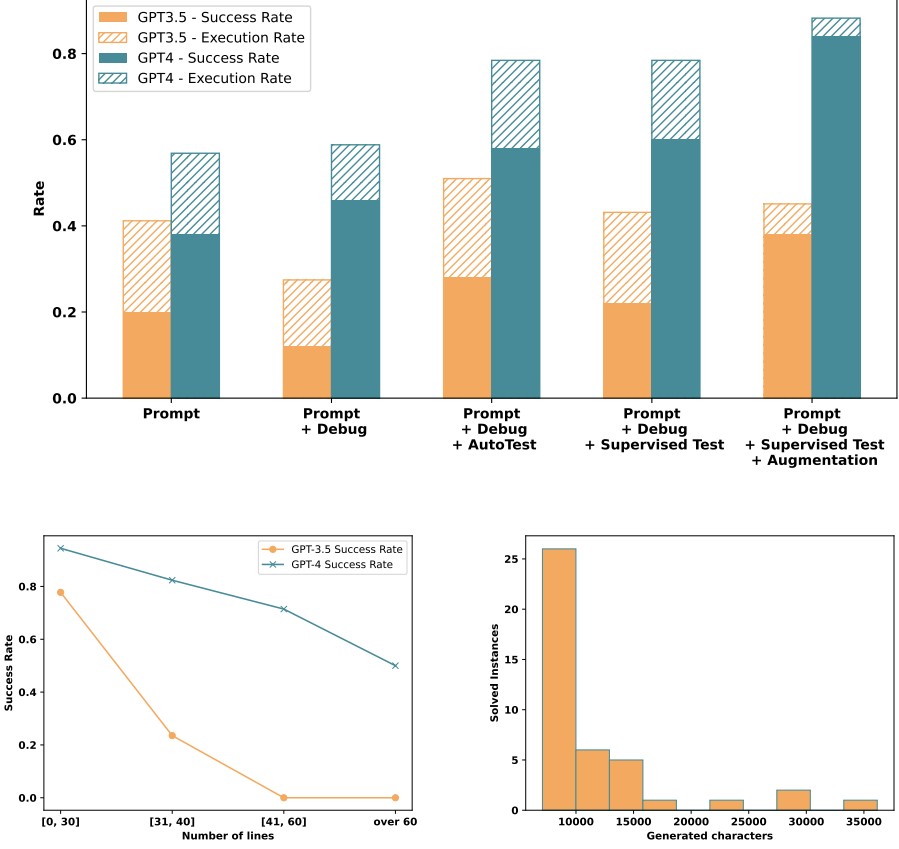

Figure 6: OptiMUS outperforms standard prompting by a large margin. Top: A comparison of the success rate and the execution rate for different modes on NLP4LP dataset using GPT-3.5 and GPT-4. Bottom left: Performance of OptiMUS vs the length of the SNOP. Bottom right: Distribution of the number of generated characters for solved instances using OptiMUS and GPT4

- Prompt: The problem is described in a few prompts and the LLM is asked to formulate the problem and write code to solve it using a given optimization solver. The code is run and the output is observed. We use this mode as a baseline.

- Prompt + Debug: In addition to the above, if the code raises syntax or runtime errors, the errors along with the code and the problem info are passed to the language model, which is prompted to debug the code. This cycle is repeated until the code runs or the maximum iterations are reached.

- Prompt + Debug + AutoTests: In addition to the above, when the code successfully runs and produces an output, automatically-generated tests are run to check the validity of the output. If the tests fail, the error messages are included with the problem description and passed back to the LLM for revision until the tests pass or the maximum iterations are reached.

- Prompt + Debug + Supervised Tests: Same as the above, except that the automatically-generated tests are all manually revised and fixed by experts if necessary.

- Prompt + Debug + Supervised Tests + Augmentation (OptiMUS): In addition to the above, each problem is rephrased using an LLM five times, and then the whole pipeline is applied to each of the rephrased versions independently. The problem is solved if at least one of the rephrased versions is solved.

We assess the models based on two metrics: success rate (the ratio of outputs passing the ground-truth supervised tests, i.e. satisfying all constraints and finding the optimal solution), and execution rate (the ratio of generated codes that are executable and generate an output). The results are available in Fig. 6.

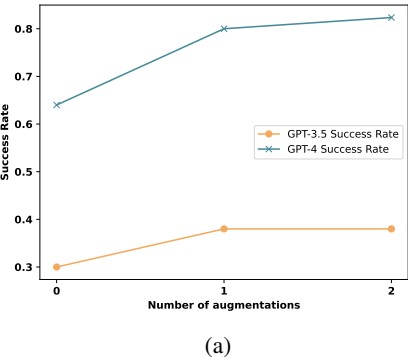 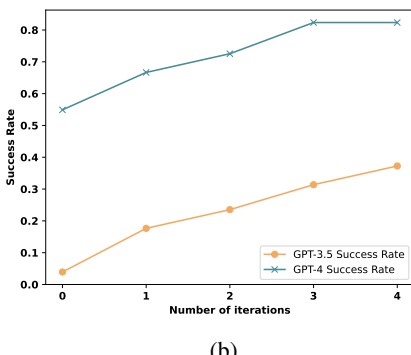

(a)                                                                  (b)

Figure 7: Comparison of the average success rate of OptiMUS with GPT-3.5 and GPT-4 on NLP4LP instance for (a) different number of augmentations and (b) different number of iterations (each iteration consists of passing an erroneous code to the LLM, prompting it to fix it, and running the revised code)

Using GPT-4, the success rate of our method improves with each of the additional features and capabilities described above. Basic prompting yields the lowest performance, as anticipated. Debugging improves the model's ability to correct execution errors, while increasing the execution rate. Adding AutoTests improves the success rate as it helps the model identify errors not found during the execution stage. Expert supervision to fix tests enhances performance but lowers the execution rate, as some revisions to fix test errors may inadvertently break the code. Finally, adding augmentations improves both success and execution rates, as OptiMUS can approach and solve problems in different ways, increasing the probability of finding a correct solution.

We observe a slightly different trend for GPT-3.5. Debugging improves the execution rate to some degree, but automated tests decrease performance due to the generation of incorrect, misleading tests. Compared to GPT-4, generating incorrect tests is more common when using GPT-3.5 since it is a weaker model. Supervised test-fixing brings performance back to the level of simple prompting, showing that GPT-3.5 is not capable of correcting codes based on test error messages. However, augmentations significantly improve the success rate by giving the model multiple attempts to solve the problem with different rephrasings.

More augmentations can improve the performance of OptiMUS at the cost of more computation. We observe that adding one or two augmentations increases the performance considerably, but additional augmentations beyond that point almost does not change OptiMUS's performance anymore (See Fig. 7a). The reason is that further rephrasings after that point result in similar outputs to the initial ones.

For GPT-4, increasing the number of iterations up to 3 can improve the performance, but going beyond that point is not useful (See Fig. 7b). If the model is not able to solve the problem in its first few attempts, it usually gets stuck in an incorrect solution space and is often not able to find the right solution. For GPT-3.5, we see that the performance keeps going up. The reason is that GPT-3.5 makes simple mistakes more frequently, and it can fix them if it is given the chance to do so. Debugging step can help with fixing minor errors like missing constraints or syntax/semantic errors, but it is usually not useful for fixing more fundamental errors like incorrect modeling problems.

Successful runs of the agent generate $4117.1 \pm 1509.7$ tokens on average. Given the fact that we used API calls to generate tokens, the formulation speed depends on factors like the internet speed, server responsiveness, model size, account priority, etc. In our experiments, all of the runs took less than 7 minutes (this is for all formulation and code/test generation steps, and does not include the solver run time). Note that OptiMUS can tackle augmented instances of the same problem in parallel.

## 6 RELATED WORK

Many authors have considered the use of LLMs to solve mathematical problems. Most of the work in this domain is focused on training and tuning models on new datasets. Frieder et al. (2023) introduce two datasets of math problems in algebra, counting, proof, calculus, probability, and various other topics to evaluate the performance of ChatGPT (OpenAI, 2023). Yuan et al. (2023) propose an arithmetic dataset, MATH 401, to evaluate the performance of LLMs on arithmetic operations. Lewkowycz et al. (2022) further trains PaLM on a dataset of mathematical and scientific papers taken from arxiv. These papers aim to evaluate and improve the direct performance of LLM, but do not seek to integrate LLMs with other tools.

Other recent studies have explored ways to improve the accuracy and reach of LLMs by improving prompts and connecting LLMs to external tools. Wei et al. (2023) propose prompting the model to consider a series of intermediate reasoning steps to improve the performance of LLMs. They also use a calculator as an external tool to execute mathematical operations for LLMs. Gao et al. (2022) uses an LLM to read natural language problems and generate programs as the intermediate reasoning steps, but offloads the solution step to a runtime such as a `Python` interpreter. He-Yueya et al. (2023) improve the performance of LLMs by delegating expression calculations to symbolic solvers. These methods take a step forward in augmenting LLMs with other tools.

However, these models are general-purpose and aim at a wide range of tasks. In contrast, our work exploits the particular structure of optimization problems to develop a new workflow for optimization modeling using LLMs together with optimization solvers that improves on simple prompting of LLMs.

Many research efforts aim to improve existing optimization solvers and develop new ones, using exact algorithms, heuristics, local and global search, simplex methods, branch and bound, dynamic programming, simulated annealing, and other methods (Adby, 2013; Koziel & Yang, 2011). Deep learning has also been used to enhance optimization solvers (Bengio et al., 2021; Cappart et al., 2021; Mazyavkina et al., 2021). Recent work has explored the possibility of using LLMs directly as solvers (Yang et al., 2023). In addition, ongoing work to enhance large language models (LLMs) (OpenAI, 2023; Touvron et al., 2023; Hoffmann et al., 2022; Vaswani et al., 2017) can support and enhance our results as the underlying LLMs improve.

Before the emergence of LLMs, there also exist work from constraint programming that design systems on extracting constraints automatically from a given library Beldiceanu & Simonis (2012); Bessiere et al. (2017). These systems are stable and accurate. However, unlike LLMs, they are unable to deal with constraints beyond the pre-specified library.

## 7 CONCLUSION

In summary, we developed OptiMUS, a Large Language Model (LLM)-based agent designed to formulate and solve optimization problems interpreted from natural language. We constructed NLP4LP, a novel dataset for optimization modeling from natural language, utilizing it to demonstrate the efficacy of the techniques implemented within OptiMUS. Our research serves as a proof of concept, illustrating the potential for automating various stages of the optimization procedure by leveraging LLMs together with traditional solvers.

Several avenues remain unexplored and can be further investigated. The quality of prompts can be enhanced through methods such as those proposed in Yang et al. (2023). The NLP4LP dataset can be expanded by adding more instances and including other classes of optimization problems. It would be interesting to enable OptiMUS to work with unstructured natural language representations of problems instead of SNOPs. Moreover, performance of OptiMUS can be potentially improved by fine-tuning the LLM specifically for optimization problem modeling and solving.

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

# A  APPENDIX

```json
{
    "allocated": [allocated_i for i in 1, ..., O],
    "price": [price_p for p in 1, ..., P],
    "input": [[input_l_i for i in 1, ..., O] for l in 1, ..., L],
    "output": [[output_l_p for p in 1, ..., P] for l in 1, ..., L],
    "cost": [cost_l for l in 1, ..., L]
}
```

```json
{
    "allocated": [8000, 5000],
    "price": [38, 33],
    "input": [
        [3, 5],
        [1, 1],
        [5, 3]],
    "output": [
        [4, 3],
        [1, 1],
        [3, 4]],
    "cost": [51, 11, 40]
}
```

```json
{
    "revenue": revenue,
    "execute": [execute_l for l in 1, ..., L]
}
```

```json
{
    "revenue": 339000.0,
    "execute":[0.0, 500.0, 1500.0]
}
```

Figure 8: An example of input (top left) and output (bottom left) formats expressed in a SNOP and their equivalent data.json (top right) and output.json (bottom right) files.

# B  SEPARATE SOLVE RATES FOR MILPS AND LPS

```
Your task is to formulate and solve the given optimization problem as a
{PROBLEM_TYPE}. Please read the problem information, input format, and objective
carefully and provide a detailed mathematical formulation.

### PROBLEM INFORMATION:

{PROBLEM_INFO}

### INPUT FORMAT:

{INPUT_FORMAT}

- Variables enclosed in [ ] represent lists of values.
- Names enclosed in quotes (") represent keys and are identical to those in the data
file.
- All other items are variables as described in the problem description and should
be replaced with their actual values from the data file.

### OBJECTIVE:

{OBJECTIVE}

### OUTPUT INFORMATION:

{OUTPUT_INFO}

### OUTPUT FORMAT:

{OUTPUT_FORMAT}

### INSTRUCTIONS:
1. Clearly define the decision variables.
2. Formulate the objective function precisely.
3. List all the constraints, ensuring they are complete and non-redundant.
4. Ensure the formulation is coherent, logical, and solvable.
5. Provide any necessary explanations or clarifications for your formulation.

Please respond with a well-structured mathematical formulation of the given
optimization problem, adhering to the instructions and format provided above.
```

Figure 9: Template for formulating the optimization problems

```
Now, please generate Python code using {SOLVER} to solve the formulated optimization
problem.
At this step, generate Python code to create the optimization variables for the problem.
These variables should be defined based on the problem data read from "data.json".

### INSTRUCTIONS:
1. **Read Data**: Read the necessary data from "data.json" (keys are strings).
2. **Variable Creation**: Define the variables based on the problem data, specifying their
type (e.g., continuous, integer, binary), and any specific bounds or constraints.
3. **Code Structure**: Structure your code clearly for variable creation. Ensure that you
use the appropriate syntax for defining variables using {SOLVER}.
4. **Solver Instructions**: {SOLVER_INSTRUCTION}
5. **Libraries**: Do not include the installation of libraries; assume all necessary
libraries are already installed.
6. **Markdown**: Wrap the generated code using markdown triple backticks (```).

When writing the code, remember that

- all keys that you read from the data file are strings and not integers
- ONLY generate the code, and don't generate anything else!

### Example
Here's an example of declaring variables with {SOLVER}:

```python
{SOLVER_VAR_DEMO}
```

Take a deep breath and work on this problem step by step. Only generate codes relevant to
variables, and no comment is needed.
```

Figure 10: Template for generating code (variables) for the optimization problems

```
Now, please generate Python code using {SOLVER} to solve the formulated optimization
problem.
At this step, generate Python code to create the optimization variables for the problem.
These variables should be defined based on the problem data read from "data.json".

### INSTRUCTIONS:
1. **Read Data**: Read the necessary data from "data.json" (keys are strings).
2. **Variable Creation**: Define the variables based on the problem data, specifying their
type (e.g., continuous, integer, binary), and any specific bounds or constraints.
3. **Code Structure**: Structure your code clearly for variable creation. Ensure that you
use the appropriate syntax for defining variables using {SOLVER}.
4. **Solver Instructions**: {SOLVER_INSTRUCTION}
5. **Libraries**: Do not include the installation of libraries; assume all necessary
libraries are already installed.
6. **Markdown**: Wrap the generated code using markdown triple backticks (```).

When writing the code, remember that

- all keys that you read from the data file are strings and not integers
- ONLY generate the code, and don't generate anything else!

### Example
Here's an example of declaring variables with {SOLVER}:

```python
{SOLVER_VAR_DEMO}
```

Take a deep breath and work on this problem step by step. Only generate codes relevant to
variables, and no comment is needed.
```

Figure 11: Template for generating code (constraints) for the optimization problems

```
Now, please generate Python code using {SOLVER} to solve the formulated optimization
problem.
At this step, generate Python code to create the optimization variables for the problem.
These variables should be defined based on the problem data read from "data.json".

### INSTRUCTIONS:
1. **Read Data**: Read the necessary data from "data.json" (keys are strings).
2. **Variable Creation**: Define the variables based on the problem data, specifying their
type (e.g., continuous, integer, binary), and any specific bounds or constraints.
3. **Code Structure**: Structure your code clearly for variable creation. Ensure that you
use the appropriate syntax for defining variables using {SOLVER}.
4. **Solver Instructions**: {SOLVER_INSTRUCTION}
5. **Libraries**: Do not include the installation of libraries; assume all necessary
libraries are already installed.
6. **Markdown**: Wrap the generated code using markdown triple backticks (```).

When writing the code, remember that

- all keys that you read from the data file are strings and not integers
- ONLY generate the code, and don't generate anything else!

### Example
Here's an example of declaring variables with {SOLVER}:

```python
{SOLVER_VAR_DEMO}
```

Take a deep breath and work on this problem step by step. Only generate codes relevant to
variables, and no comment is needed.
```

Figure 12: Template for generating code (objective) for the optimization problems

```
Given an optimization problem with the following details:

### PROBLEM TYPE:
 {PROBLEM_TYPE}

### PROBLEM INFO:
 {PROBLEM_INFO}

### INPUT FORMAT:
 {INPUT_FORMAT}

### OBJECTIVE:
 {OBJECTIVE}

### OUTPUT INFO:
 {OUTPUT_INFO}

### OUTPUT FORMAT:
 {OUTPUT_FORMAT}

A code reads the input from file called `data.json`, solves the problem, and writes
the output in an `output.json` file. Develop a script that validates the generated
output by checking for constraint violations and making sure the json format (keys)
of the output is correct. I have a template for the test already:

{INITIAL_TEST_SCRIPT}

Complete the `run()` function of the script to perform the required validation
checks as per the above requirements.

### **Requirements:**
1. **Checks:** Generate a bullet list of checks the script should perform.
2. **Script Modification:** Only modify the `run()` function in the provided initial
test script.
3. **Return Value:** The `run()` function should return a list of strings indicating
the errors in the output, or an empty list ([]) if the output is valid.
4. **Libraries:** Assume necessary libraries are already installed.
5. **Numerical Inaccuracy:** Use `eps` for all numerical comparisons (==, <=, <, >=,
>) to account for any numerical inaccuracies. For instance, use x > -eps instead of
x > 0.
6. **Error Messages:** Generate informative and descriptive error messages, using f-
strings where appropriate, to convey what is wrong with the output.

### **Instructions:**
- Start with imports and generate the complete script.
- All keys read from the data file are strings and not integers.
- Wrap the code in markdown (```).

### **Task:**
Complete the `run()` function of the script to perform the required validation
checks as per the above requirements.
```

Figure 13: Template for generating tests for the optimization problems

```
I will provide you with sentences that describe an optimization problem. Your task
is to rephrase the problem to ensure clarity and precision while maintaining the
integrity of the mathematical and technical aspects of the problem.

{PROBLEM_INFO}

- Please use "-"s at the beginning of each statement.
- Do not alter items inside \var wrappers.
- Aim for a rephrasing that is distinct in structure and wording but preserves the
original meaning and details of the problem.
```

Figure 14: Template for rephrasing the optimization problems

```
I am trying to solve the problem using a {SOLVER} code in python. The code reads
data from a file named "data.json" and saves the output in a file named
"output.json". When I run this code I get some errors.

### ORIGINAL CODE:
{CODE}

### ERROR MESSAGES:

{ERROR_MESSAGE}

### INSTRUCTIONS:

- Review the code carefully, considering the provided context.
- Identify and correct any syntax errors in the code.
- Ensure that the corrected code runs without errors and produces the expected
output.
- Wrap the generated code using markdown triple backticks (```) to maintain format.

### YOUR TASK:

- First read the code and understand what the problem(s) exactly are.
- Then provide a corrected version of the code that resolves the errors and
generates the expected outcome.
```

Figure 15: Template for debugging the code based on execution errors

```
I am using a {SOLVER} code in python to formulate and solve this problem. The code,
provided below, reads data from a file named "data.json" and writes the output to a
file named "output.json". However, a user has reported that the output is incorrect
and provided a list of error messages.

### CODE:

{CODE}

### ERROR MESSAGE:

{ERROR_MESSAGE}

### YOUR TASK:

1. Could you please provide a detailed explanation of each error mentioned in the
{ERROR_MESSAGE}?
2. Could you identify the sections of the code that are likely causing these errors?
3. Could you suggest modifications to the code to resolve the errors and fix the
output?

Based on the above information, please generate a corrected version of the entire
code.
```

Figure 16: Template for fixing the code based on test errors

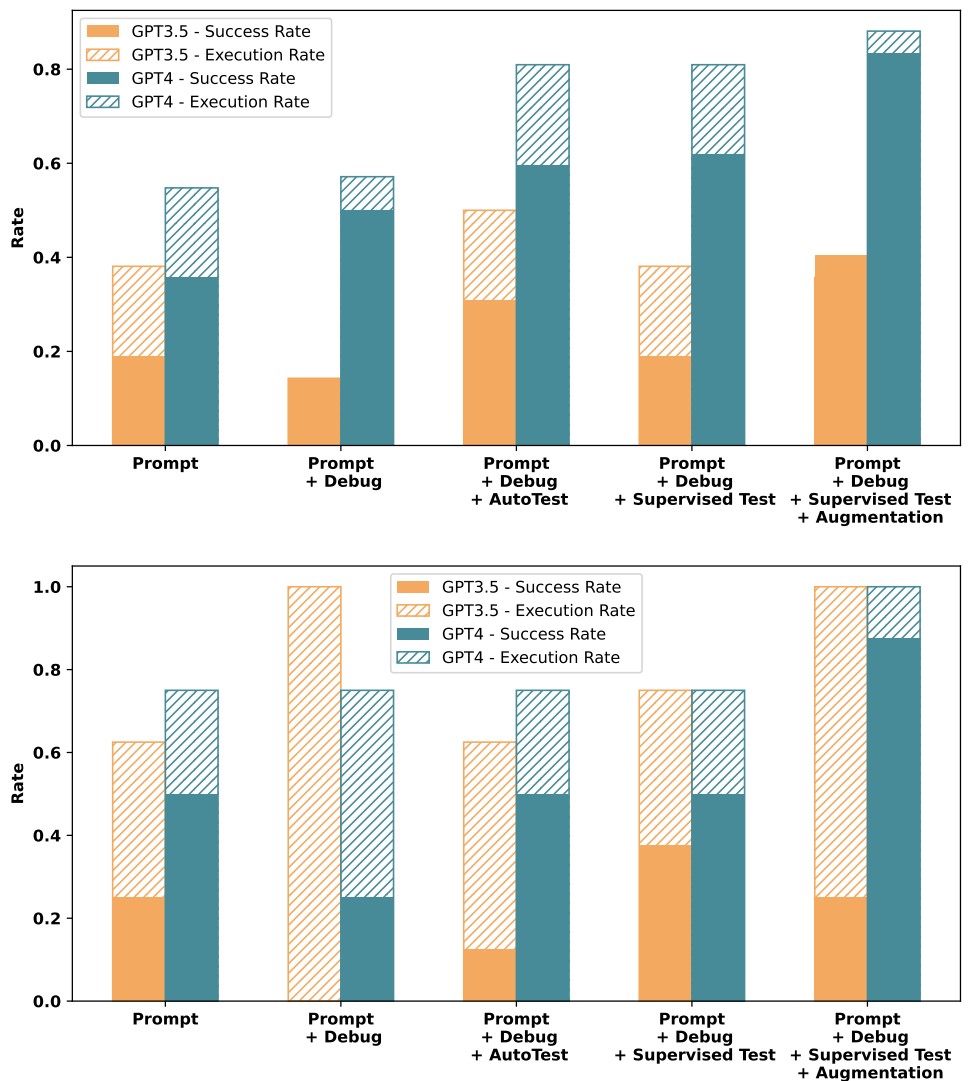

Figure 17: Separate figures for LPs and MILPs. Above: LP; Below: MILP

