# OptiMUS: Optimization Modeling Using mip Solvers and large language models

## Abstract

Optimization problems are pervasive across various sectors, from manufacturing and distribution to healthcare. However, most such problems are still solved heuristically by hand rather than optimally by state-of-the-art solvers, as the expertise required to formulate and solve these problems limits the widespread adoption of optimization tools and techniques. We introduce OptiMUS, a Large Language Model (LLM)-based agent designed to formulate and solve MLIP problems from their natural language descriptions. OptiMUS is capable of developing mathematical models, writing and debugging solver code, developing tests, and checking the validity of generated solutions. To benchmark our agent, we present NLP4LP, a novel dataset of linear programming (LP) and mixed integer linear programming (MILP) problems. Our experiments demonstrate that OptiMUS is able to solve 67% more problems compared to a basic LLM prompting strategy. The code OptiMUS and the data for NLP4LP are available at https://anonymous.4open.science/r/nlp4lp-8F62/README.md

## 1 Introduction

Optimization problems are common in many fields like operations, economics, engineering, and computer science. Reducing energy use in smart grids, improving supply chains, or increasing profits in algorithmic trading are only a few applications of optimization (Singh, 2012; Antoniou & Lu, 2007). There are various types of optimization problems, such as linear programming (LP) and mixed integer linear programming (MILP), each with their own algorithms and constraints. However, the success of any solution depends on both the choice of algorithm and how the problem is modeled, since different models of the same problem can lead to different performances. This creates a challenge: one needs expert knowledge to both formulate and solve these problems. This expertise gap makes it hard for many sectors to take advantage of optimization, even though it could significantly improve their operations. Examples

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

Given an output, we need to ensure its correctness. OptiMUS generates unit tests using the SNOP to ensure the validity of the output. These tests check for correct formatting and constraint satisfaction. See Fig. 4 for an example. Optionally, a user of OptiMUS can also revise these generated tests or write additional tests. We call these auto-generated tests, supervised tests, and human tests, respectively. Our benchmark includes supervised tests for every problem. In our experience, developing supervised tests is roughly five times faster than developing equivalent human tests from scratch.

Given an output, OptiMUS runs the unit tests. Any tests that fail will generate error messages. For example, the tests might detect negative order quantities or inconsistencies within monthly revenue, cost, and profit values.

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

OUTPUT INFO:
{OUTPUT_INFO}

your code should save the output as a file named "output.json" with indents of 4
spaces in this format:

{OUTPUT_FORMAT}

when writing the code, remember that
{SOLVER_INSTRUCTION}
- all keys that you read from the data file are strings and not integers
- try to keep the code for adding different constraints separate to avoid confusion
- ONLY generate the code, and don't generate anything else! I've already installed
the necessary libraries.
- Wrap the code using markdown ```

### INSTRUCTIONS:
1. **Read Data**: Read all necessary data from "data.json". Remember, all keys in
the data file are strings.
2. **Solver Instructions**:
{SOLVER_INSTRUCTION}
3. **Code Structure**: Structure your code clearly, keeping the code for adding
different constraints separate to maintain readability and avoid confusion.
4. **Output File**: Save the results in "output.json" using the specified format.
5. **Libraries**: Do not include installation of libraries; assume all necessary
libraries are already installed.
6. **Markdown**: Wrap the generated code using markdown triple backticks (```) to
maintain format.
```

Figure 9: Template for generating code for the optimization problems

```
Given an optimization problem with the following details:

### PROBLEM TYPE:
 {PROBLEM_TYPE}

### PROBLEM INFO:
 {PROBLEM_INFO}

### INPUT FORMAT:
 {INPUT_FORMAT}

### OBJECTIVE:
 {OBJECTIVE}

### OUTPUT INFO:
 {OUTPUT_INFO}

### OUTPUT FORMAT:
 {OUTPUT_FORMAT}

A code reads the input from file called `data.json`, solves the problem, and writes
the output in an `output.json` file. Develop a script that validates the generated
output by checking for constraint violations and making sure the json format (keys)
of the output is correct. I have a template for the test already:

{INITIAL_TEST_SCRIPT}

Complete the `run()` function of the script to perform the required validation
checks as per the above requirements.

### **Requirements:**
1. **Checks:** Generate a bullet list of checks the script should perform.
2. **Script Modification:** Only modify the `run()` function in the provided initial
test script.
3. **Return Value:** The `run()` function should return a list of strings indicating
the errors in the output, or an empty list ([]) if the output is valid.
4. **Libraries:** Assume necessary libraries are already installed.
5. **Numerical Inaccuracy:** Use `eps` for all numerical comparisons (==, <=, <, >=,
>) to account for any numerical inaccuracies. For instance, use x > -eps instead of
x > 0.
6. **Error Messages:** Generate informative and descriptive error messages, using f-
strings where appropriate, to convey what is wrong with the output.

### **Instructions:**
- Start with imports and generate the complete script.
- All keys read from the data file are strings and not integers.
- Wrap the code in markdown (```).

### **Task:**
Complete the `run()` function of the script to perform the required validation
checks as per the above requirements.
```

Figure 10: Template for generating tests for the optimization problems

```
I will provide you with sentences that describe an optimization problem. Your task
is to rephrase the problem to ensure clarity and precision while maintaining the
integrity of the mathematical and technical aspects of the problem.

{PROBLEM_INFO}

- Please use "-"s at the beginning of each statement.
- Do not alter items inside \var wrappers.
- Aim for a rephrasing that is distinct in structure and wording but preserves the
original meaning and details of the problem.
```

Figure 11: Template for rephrasing the optimization problems

```
I am trying to solve the problem using a {SOLVER} code in python. The code reads
data from a file named "data.json" and saves the output in a file named
"output.json". When I run this code I get some errors.

### ORIGINAL CODE:
{CODE}

### ERROR MESSAGES:

{ERROR_MESSAGE}

### INSTRUCTIONS:

- Review the code carefully, considering the provided context.
- Identify and correct any syntax errors in the code.
- Ensure that the corrected code runs without errors and produces the expected
output.
- Wrap the generated code using markdown triple backticks (```) to maintain format.

### YOUR TASK:

- First read the code and understand what the problem(s) exactly are.
- Then provide a corrected version of the code that resolves the errors and
generates the expected outcome.
```

Figure 12: Template for debugging the code based on execution errors

```
I am using a {SOLVER} code in python to formulate and solve this problem. The code,
provided below, reads data from a file named "data.json" and writes the output to a
file named "output.json". However, a user has reported that the output is incorrect
and provided a list of error messages.

### CODE:

{CODE}

### ERROR MESSAGE:

{ERROR_MESSAGE}

### YOUR TASK:

1. Could you please provide a detailed explanation of each error mentioned in the
{ERROR_MESSAGE}?
2. Could you identify the sections of the code that are likely causing these errors?
3. Could you suggest modifications to the code to resolve the errors and fix the
output?

Based on the above information, please generate a corrected version of the entire
code.
```

Figure 13: Template for fixing the code based on test errors