# OpenReview forum: "OptiMUS: Optimization Modeling Using mip Solvers and large language models"
_ICLR.cc/2024/Conference — Submitted to ICLR 2024_

### Official Review · Reviewer_8JDZ · 2023-10-26

**Soundness:** 1 poor
**Presentation:** 1 poor
**Contribution:** 2 fair
**Rating:** 3
**Confidence:** 4

**Summary:**

This introduction highlights the wide-ranging prevalence of optimization problems in operations, economics, engineering, and computer science. It emphasizes their significance in applications like improving energy efficiency in smart grids, refining supply chains, and enhancing profits in algorithmic trading. The paper underlines the critical role of algorithm selection and problem modeling in achieving successful solutions.
The expertise required to navigate these challenges creates a barrier for many sectors, including supermarkets, hospitals, municipalities, solar farms, and small businesses, limiting their access to optimization benefits.
The paper proposes leveraging Large Language Models (LLMs) to democratize access to optimization. LLMs have demonstrated proficiency in understanding and generating natural language, offering a means to simplify problem formulation and disseminate expert knowledge. However, their role in optimization remains underexplored due to their novelty and the lack of comprehensive benchmarks.
The paper introduces three key contributions:
1. The NLP4LP dataset comprises 40 expert-formulated linear programming (LP) and mixed integer linear programming (MILP) problems. It includes annotated solutions, optimality-checking code, and sample formulations in markdown and code formats. The dataset uses a standardized format for representing optimization problems in natural language.
2. OptiMUS, an LLM-based agent designed for formulating and solving optimization problems, was introduced.
3. Developing techniques to enhance OptiMUS's performance, including automated data augmentation through problem rephrasing and self-improvement of solutions via automated testing and debugging. These techniques lead to a 67% increase in the solve rate compared to direct prompting.
In summary, the paper's contributions aim to democratize access to optimization techniques across various domains, extending their reach and utility.

**Strengths:**

The strengths of this paper are as follows.
1. Proposed a new framework for solving optimization problems using natural language.
2. Proposed SNOP, a new format for expressing optimization problems in natural language.
3. This paper proposed NLP4LP, a new dataset of optimization problems expressed in SNOP.

**Weaknesses:**

The final sentence of the first paragraph on page 4 and the enumeration at the beginning of page 7.

Weaknesses regarding the content of the text are as follows:
3. The results of solving 32 different LPs and 8 different MILPs are summed for the experiment. This bais of problem types makes it difficult to compare whether the results of the experiments are more influenced by the nature of the problem (LP or MILP) or the nature of the method.
4. The experiment in Figure 6 should describe the problems used. It is difficult to determine whether the results are the average of multiple problems or the results of solving one problem.
5. There is no description of the solver used in the experiments. We believe that the choice of solver is important to improve the success rate and execution rate. Therefore, the experiment section should state what solver was used and, if OptiMUS selected it, how it was selected.

This study covers an exciting subject. I hope that future studies will improve the weaknesses mentioned above.

**Questions:**

In addition to the above weaknesses, I would like to have the following questions answered:
1. Is there any difference between different types of problems for variation in success rate and execution rate; please tell us about the experiment results in Figure 5, focusing only on LP (MILP).
2. What is the maximum number of iterations with debugging?

---

> ### Author Response · Authors · 2023-11-18
> **Response to reviewer 8JDZ**
>
> **Response to reviewer 8JDZ**
>
> We thank the reviewer for the time spent on our paper and valuable suggestions.
>
> ----
> **Weaknesses**
>
> 1. Reviewer’s concern: This bias of problem types makes it difficult to compare whether the results of the experiments are more influenced by the nature of the problem (LP or MILP) or the nature of the method
>
>    We thank the reviewer for pointing this out. We have included plots for the performance of OptiMus on LP and MILP separately (Please see appendix B in the revised paper). We observe that OptiMUS performs the best among all the methods we evaluated in both cases, solving more than 80% of the problems.
>
> 2. Reviewer’s concern: The experiment in Figure 6 should describe the problems used.
>
>    Sorry for the confusion. Figure 6 is plotted based on the whole dataset. We added more details to the figure caption.
>
> 3. Reviewer’s concern: There is no description of the solver used in the experiments
>
>    We used Gurobi for all our tests. The reason for this choice is that we noticed that there are abundant internet resources on the Gurobi Python interface. Moreover, according to our preliminary tests, Gurobi performs better than CVXPY in general. We also updated the experiment section in the revised version to include this information.
>
> 4. Reviewer’s concern: typos and formatting issues
>
>    Thanks for pointing them out. We have fixed the issues in our revision.
>
> ----
> **Questions**
>
> 1. Reviewer’s question: Is there any difference between different types of problems for variation in the metric?
>
>    Please refer to our response to concern 1 in the weaknesses section.
>
> 2. What is the maximum number of iterations with debugging
>
>    The maximum number of iterations means the maximum number of times OptiMUS is allowed to debug the code if an execution error or a test error is raised. We use 5 debugging iterations in our experiments. We added more details on this to the revised version.
>
> We sincerely appreciate the valuable suggestions of the reviewer. We have addressed all of the reviewer’s concerns. If the reviewer finds our response and revision satisfactory, we politely request the reviewer to update the score accordingly.

---

> > ### Comment · Reviewer_8JDZ · 2023-11-22
> >
> > Thank you for your courteous response.
> >
> > I am an expert in optimization problems such as LP (Linear Programming) and MILP (Mixed Integer Linear Programming). However, LP and MILP differ in both algorithms and computational complexity. For instance, in MILP, simply changing the order of variables and constraints can significantly alter the performance of solvers. As an expert in optimization problems, I am very interested in how much this method can mitigate computational difficulties, but regarding MILP, considering its NP-hardness, the outcome seems negative.
> >
> > Of course, I do not intend to deny the results of this experiment. However, since it is impossible to judge whether the results are limited to certain instances, it feels premature to conclude the effectiveness of this method in LP and MILP.

---

> > > ### Author Response · Authors · 2023-11-22
> > > **Thank you for your response**
> > >
> > > We thank the reviewer for the reply. We agree that it's a very interesting direction to see if LLMs improve the computational aspects of LP/MILP (such as performance variability mentioned by the reviewer).
> > >
> > > However, we need to make some clarifications:
> > >
> > > 1. Our paper focuses on the modeling layer of LP/MILP, instead of solving them numerically. LLMs are applied to transform natural language description of a problem into formulation and executable solver code. After building the model, we call solvers to solve these models. This procedure involves logic reasoning, rather than the algorithmic or computational aspects. Actually, as we mentioned in **Novelty** of our global response, one of our major contributions is to avoid passing numerical data to LLMs.
> > >
> > > 2. In terms of benchmarking, we politely disagree with saying "it is impossible to judge whether the results are limited to certain instances". In terms of measuring MILP/LP softwares, it is common to test them on a large set of benchmark instances. And it is possible to use quantitative metrics like SGM to evaluate a numerical software.
> > > However, when it comes to modeling, we care about *whether the model is a true reflection of the underlying problem*. Our dataset contains a variety of optimization problems (routing, scheduling, resource allocation, supply chain, combinatorial,...), and we believe this already demonstrates the efficacy of LLMs in the modeling task.
> > >
> > > ----
> > > In summary, we believe that LLMs can help making solvers more accessible to the public by simplifying the modeling/coding process.
> > >
> > > ----
> > > Thank you again for the time spent reviewing our paper. And we hope that our response addresses the reviewer's concern.

---

> > > > ### Comment · Reviewer_8JDZ · 2023-11-22
> > > >
> > > > Thanks for the quick reply.
> > > > Since it is not feasible to experiment with an infinite number of instances, using a benchmark set is naturally the course of action. The question is whether this benchmark set is considered sufficient in its type, quantity, and difficulty level in that field. For this reason, benchmark sets like MIPLIB have been proposed. For example, in the case of MILP, I believe that it cannot be considered sufficient with as few as eight instances.
> > > > > 2. In terms of benchmarking, we politely disagree with saying "it is impossible to judge whether the results are limited to certain instances". In terms of measuring MILP/LP softwares, it is common to test them on a large set of benchmark instances.

---

> > > > > ### Author Response · Authors · 2023-11-22
> > > > > **Thank you for your response**
> > > > >
> > > > > We again appreciate the reviewer's prompt reply and comments regarding the dataset.
> > > > >
> > > > > We acknowledge that there are fewer publicly available MILP models compared to LP models. However, although our MIP dataset contains 8 instances, it still covers various problem types and exhibits high diversity. With our augmentation technique, each problem is expanded to 6 unique problems (including the original and 5 augmented versions), overall giving 48 testing problems in our experiment. Our experiments suggest OptiMUS obtains similar performance improvement on MILP as in the LP case.
> > > > >
> > > > > Last we would like to emphasize that, while MILP and LP differ a lot in terms of computational algorithms, they exhibit similar nature when it comes to modeling. We believe that our experiment already demonstrates that LLMs have the potential to handle the modeling tasks, so as to make optimization more accessible to the public.
> > > > >
> > > > > Here's a summary of problem types of our MILPs
> > > > >
> > > > > | Problem | Type                    |
> > > > > | ------- | ----------------------- |
> > > > > | 1       | Supply chain management |
> > > > > | 2       | Unit commitment         |
> > > > > | 3       | Facility location       |
> > > > > | 4       | Mining                  |
> > > > > | 5       | Production planning     |
> > > > > | 6       | Crew scheduling         |
> > > > > | 7       | Combinatorial           |
> > > > > | 8       | Crew scheduling     |

---

> > > > > > ### Comment · Reviewer_8JDZ · 2023-11-23
> > > > > >
> > > > > > I would like to add to my earlier comments.
> > > > > > It's not mandatory to use MIPLIB, but I strongly suggest referring to the MIPLIB homepage and papers at https://miplib.zib.de/.
> > > > > >
> > > > > > In the world of deep learning, researchers debate over appropriate datasets like ImageNet and CIFAR.
> > > > > > Similarly, in the world of MIP (Mixed Integer Programming), there has been a long-standing discussion about what kind of datasets are necessary. It's not just about categorizing problems; factors like the type of formulation and problem size are also carefully considered in the selection of problems.
> > > > > >
> > > > > > It's fine to prepare a new dataset, but when claiming its generality, I think careful consideration should also be given to discussions in the optimization field, similar to those surrounding MIPLIB.

---

> > > > > > > ### Author Response · Authors · 2023-11-23
> > > > > > > **Thank you for your reply**
> > > > > > >
> > > > > > > We thank the reviewer for the prompt response.
> > > > > > >
> > > > > > > MIPLIB has been a long standing dataset to test the computational efficiency of MIP solvers. But when it comes to testing a model, we believe that a different methodology is needed.
> > > > > > >
> > > > > > > MIPLIB contains realizations of optimization models, where for each instance we can define problem size, sparsity, constraint type, and other metrics/features. But in terms of a mathematical model, we cannot quantify its features, nor can we let LLM read large mps files (e.g., even for a simple knapsack problem its mps may be gigabytes long). This motivates us to collect maths models in natural language. Moreover, we hope to remind the reviewer of our major contribution: creating an LLM agent to help building mathematical optimization models.
> > > > > > >
> > > > > > > We indeed find the reviewer's perspective constructive in further augmenting our dataset, and we agree that it's worth a careful consideration how to build up a modeling dataset for MIPs. Thank you for your efforts in the review process.

---

### Official Review · Reviewer_Bafe · 2023-10-30

**Soundness:** 3 good
**Presentation:** 3 good
**Contribution:** 3 good
**Rating:** 6
**Confidence:** 3

**Summary:**

The paper proposes a system to acquire the formal definition of an optimization
model from a natural language description using LLMs. The authors describe their
approach and evaluate it empirically.

**Strengths:**

The proposed system is very interesting and potentially makes solving technology
much more accessible.

**Weaknesses:**

The acquisition of MIP and similar types of problems from high-level
descriptions and examples of solutions has long been investigated, see for
example
Beldiceanu, N., Simonis, H. (2012). A Model Seeker: Extracting Global Constraint Models from Positive Examples. In: Milano, M. (eds) Principles and Practice of Constraint Programming. CP 2012. Lecture Notes in Computer Science, vol 7514. Springer, Berlin, Heidelberg. https://doi.org/10.1007/978-3-642-33558-7_13
This work should at least be mentioned, as it is highly relevant here.

There are multiple broken references (??) throughout the paper.

**Questions:**

How was the ground truth for experiments obtained? -- answered in rebuttal.

---

> ### Author Response · Authors · 2023-11-18
> **Response to reviewer Bafe**
>
> **Response to reviewer Bafe**
>
> We thank the reviewer for the time spent on our paper.
>
> ----
> **Weaknesses**
>
> 1. Reviewer’s concern: There are no ground truth models for the considered optimization problems; There is nothing to ensure that the modeled problem corresponds to the natural language description
>
>    Many mathematical optimization problems can be modeled in several different ways. Therefore, we believe it is not reasonable to limit the evaluation process to a single ground-truth model. Instead, OptiMUS focuses on verifying that the output solution is valid. As we discussed in Section 3.4 of the paper, we use a human-written testing script to benchmark each instance. An instance is solved *only if* the solution passes the human-written test. Essentially, if the model does not correspond to the actual problem, the tests will fail.
>
> 2. Reviewer’s concern: comparison to previous work
>
>    We thank the reviewer for pointing out this valuable branch of missing references, and we have included discussions in our revision. Compared to using LLMs to extract a mathematical model, using a structured system can bring great stability, accuracy, and explainability. But such systems are also less flexible. OptiMUS can accept any natural language input, whereas the automated constraint acquisition literature requires a user to select from a library of constraints.
>
> 3. Reviewer’s concern: Broken references
>
>    Thank you for pointing this out. We fixed it in our revision.
> ----
> **Questions**
>
> 1. Reviewer’s question: How was the ground truth for experiments obtained?
>
>    We obtain the optimal value of each instance either from textbook solution manuals or by manually modeling and solving the instance. Each instance has a supervised testing script that guarantees the validity of the solution. We added more details to section 4 (dataset) in the revised version.
>
> We hope that our response can address the concern of the reviewer, and thanks again for your constructive reviews.
>
> ----
> **References**
>
> [1] Beldiceanu, N., & Simonis, H. (2012, October). A model seeker: Extracting global constraint models from positive examples. In *International Conference on Principles and Practice of Constraint Programming* (pp. 141-157). Berlin, Heidelberg: Springer Berlin Heidelberg.

---

> > ### Comment · Reviewer_Bafe · 2023-11-20
> >
> > Thank you for the clarification. I completely agree that there are multiple ways to model a problem and this is not my concern here. Could you elaborate on what the human-written tests scripts check? Do they check the value of the objective function for the returned solution, or do they perform a "semantic" check to ensure that the solution satisfies the original problem?

---

> > > ### Author Response · Authors · 2023-11-20
> > >
> > > Of course! Thanks for your timely response.
> > > Human-written tests we use for benchmarking check the value of the objective function and also perform a semantic check. However, in practice, we will not have access to ground truth optimal values prior to solving the problem, and human-supervised tests will only conduct a semantic check.

---

> > > > ### Comment · Reviewer_Bafe · 2023-11-20
> > > >
> > > > Thanks -- what exactly does this semantic check look like? In particular, how do you ensure that the solution is actually a solution for the original problem, and how do you get the ground truth for what is a solution?

---

> > > > > ### Author Response · Authors · 2023-11-20
> > > > >
> > > > > We gathered the ground truth values for all problems either by solving them ourselves or by using the books' solution manuals.
> > > > >
> > > > > Our supervised tests (saved as `test-human.py` in each problem's directory) read in `output.json` file and check for these:
> > > > >
> > > > > 1. Format Validity: Checks whether the json file exists and its format matches the **OUTPUT FORMAT** specified in our SNOP (dimensionality, json keys, etc.)
> > > > >
> > > > > 2. Consistency: Checks whether the values in the output are consistent or not. For instance making sure $income_{net} = income_{gross} - cost$
> > > > >
> > > > > 3. Feasibility: Checks whether each constraint is violated (with some $\varepsilon$ tolerance) or not.
> > > > >
> > > > > Whenever a check fails, an error message is added to the error list. The list will then be passed back to the LLM to fix the code. For evaluation, we also read the ground truth optimal value from `obj.txt` and compare it with the value generated in `output.json`
> > > > >
> > > > > Here's an example testing script:
> > > > >
> > > > > ```
> > > > > import json
> > > > > ​
> > > > > eps = 1e-03
> > > > > ​
> > > > > ​# Function for parsing the output json file
> > > > > def parse_json(json_file, keys):
> > > > >     if isinstance(json_file, list):
> > > > >         for v in json_file:
> > > > >             parse_json(v, keys)
> > > > >     elif isinstance(json_file, dict):
> > > > >         for k, v in json_file.items():
> > > > >             if isinstance(v, dict) or isinstance(v, list):
> > > > >                 parse_json(v, keys)
> > > > >             if k not in keys:
> > > > >                 keys.append(k)
> > > > > ​
> > > > > ​
> > > > > def run():
> > > > >     with open("data.json", "r") as f:
> > > > >         data = json.load(f)
> > > > > ​
> > > > >     with open("output.json", "r") as f:
> > > > >         output = json.load(f)
> > > > > ​
> > > > >     all_json_keys = []
> > > > >     parse_json(output, all_json_keys)
> > > > > ​
> > > > >     error_list = []
> > > > >
> > > > > ​    ​# Check if the keys exist
> > > > >     if not "net_income" in all_json_keys:
> > > > >         error_list.append("The output field 'net_income' is missing")
> > > > > ​
> > > > >     if not "production" in all_json_keys:
> > > > >         error_list.append("The output field 'production' is missing")
> > > > > ​
> > > > >     if not "upgrade" in all_json_keys:
> > > > >         error_list.append("The output field 'upgrade' is missing")
> > > > > ​
> > > > >     # Check if production quantities are non-negative
> > > > >     for i, production_i in enumerate(output["production"]):
> > > > >         if production_i < -eps:
> > > > >             error_list.append(
> > > > >                 f"Production quantity of product {i+1} is negative: {production_i}"
> > > > >             )
> > > > > ​
> > > > >     # Check if machine hours do not exceed available hours
> > > > >     total_hours = sum(
> > > > >         production_i * hour_i
> > > > >         for production_i, hour_i in zip(output["production"], data["hour"])
> > > > >     )
> > > > >     if output["upgrade"]:
> > > > >         available_hours = data["availableHours"] + data["upgradeHours"]
> > > > > ​
> > > > >     if total_hours - available_hours > eps:
> > > > >         error_list.append(
> > > > >             f"Total machine hours used ({total_hours}) exceed available hours ({data['availableHours']})"
> > > > >         )
> > > > > ​
> > > > >     # Check if total cost does not exceed available cash
> > > > >     total_cost = sum(
> > > > >         production_i * cost_i
> > > > >         for production_i, cost_i in zip(output["production"], data["cost"])
> > > > >     )
> > > > > ​
> > > > >     if output["upgrade"]:
> > > > >         total_cost += data["upgradeCost"]
> > > > > ​
> > > > >     total_revenue = sum(
> > > > >         production_i * price_i * (1 - investPercentage_i)
> > > > >         for production_i, price_i, investPercentage_i in zip(
> > > > >             output["production"], data["price"], data["investPercentage"]
> > > > >         )
> > > > >     )
> > > > >     if total_cost - data["cash"] > eps:
> > > > >         error_list.append(
> > > > >             f"Total cost ({total_cost}) exceeds available cash ({data['cash']})"
> > > > >         )
> > > > > ​
> > > > >     # Check if net income is correctly calculated
> > > > >     net_income = total_revenue - total_cost
> > > > >     if abs(net_income - output["net_income"]) > eps:
> > > > >         error_list.append(
> > > > >             f"Net income is incorrectly calculated: expected {net_income}, got {output['net_income']}"
> > > > >         )
> > > > > ​
> > > > >     return error_list
> > > > > ​
> > > > > ​
> > > > > if __name__ == "__main__":
> > > > >     print(run())
> > > > > ```
> > > > >
> > > > >
> > > > > Please let us know if this answers your question.

---

> > > > > > ### Comment · Reviewer_Bafe · 2023-11-20
> > > > > >
> > > > > > Thank you, this answers my question and addresses my main concern. I'll revise my review accordingly.

---

> > > > > > > ### Author Response · Authors · 2023-11-20
> > > > > > >
> > > > > > > We are glad to hear that our responses addressed your questions and concerns. Thank you so much for your time and valuable feedback.

---

### Official Review · Reviewer_GUNL · 2023-10-31

**Soundness:** 2 fair
**Presentation:** 2 fair
**Contribution:** 3 good
**Rating:** 5
**Confidence:** 3

**Summary:**

The paper proposes using large language model (LLM) to automatize optimization modeling and solving: formulate MILP from natural language description, generate the solver code and test the output. A package OptiMUS is developed, along with a new dataset NLP4LP as benchmark set. The experiments demonstrate the potential of large language model to help model and solve optimization problems.

**Strengths:**

This work poses an intriguing question: how can large language model help optimization modeling and solving, and makes an initial exploration on this topic. The paper demonstrates the potential of LLM to assist modeling and generating solver code. Compared with direct prompting as baseline, the augmentations developed, such as iterative debugging, are demonstrated to effectively increase the execution and success rates.

**Weaknesses:**

- Though the question posed is interesting, the contribution of this work is less significant. Basically, OptiMUS leverages LLM to generate the mathematical formulation and test code from the SNOP and to iteratively debug the solver code. If so, I'm afraid this work is kind of engineering without much novelty.
- The experiments are far from exhaustive. In the paper, only execution and success rates are reported but there are many aspects of solving optimization problems to discuss. See Questions for more detail on this point.
- The instances in the benchmark set NLP4LP are collected from optimization textbooks and lecture notes. They are more like toy examples (which means they are not representative of real problems) and typically with small sizes. And conclusions drawn on 40 instances can be unreliable.

**Questions:**

Major comments: I do have several concerns on OptiMUS from the optimization solving perspective.

- If I understand correctly, the mathematical formulation of the problem is generated from the SNOP provided by users. I'm wondering how to guarantee the correctness of the mathematical formulation. Moreover, the test code is also generated by LLM. It could happen that LLM misunderstands the SNOP, generates the wrong mathematical formulation and corresponding wrong test script. In such case the solver solves a wrong problem but passes the test. Is OptiMUS able to detect and circumvent such scenarios?
- Typically, modern solvers have many parameters and options to set. Different options can generate very different outputs. Is the output of OptiMUS stable or not?
- As mentioned in the paper, OptiMUS checks correct formatting and constraint satisfaction, namely feasibility. Does OptiMUS check optimality of the output?
- The instances in NLP4LP are collected from classic optimization textbooks and lecture notes. There are chances that these materials are included in the training data of LLM, which makes the numerical results here less convincing. Moreover, these instances can be very different with real problems and they have relatively small scale. My suggestion is that experiments on real-world problems should be conducted to further demonstrate the effectiveness of OptiMUS.

Other comments:
- I find it difficult to read Figure 5. Why are success rates always higher than the execution rates? In other words, what is the definition of success rate? In Page 8, the authors write ``success rate (the ratio of outputs satisfying all constraints and finding the optimal solution)". Is it the ratio of success number and all instances, or the ratio of success number and execution count?
- In the abstract, there is a typo "MLIP" which should be "MILP".
- The references to the figures in the manuscript (without appendix) need double-check. For example, there are some missing references in page 7.
- I'm also concern about the reproducibility of the results because the output of LLM can sometimes be irreproducible. Can the authors comment on this point?

---

> ### Author Response · Authors · 2023-11-18
> **Response to reviewer GUNL**
>
> **Response to reviewer GUNL**
>
> We express our gratitude to the reviewer for their helpful feedback.
>
> ----
> **Weaknesses**
>
> 1. Reviewer’s concern: The contribution of this work is not significant enough.
>
>    We refer the reviewer to Section **Novelty** in our global response.
>
> 2. Reviewer’s concern: The experiments are not exhaustive, only execution and success rates are reported.
>
>    We conducted more experiments and included them in the revised version. We will further address the reviewer's questions in the **Questions** section of the response.
>
> 3. Reviewer’s concern: instances in the benchmark set NLP4LP are collected from optimization textbooks, and are toy examples. Conclusions drawn on 40 instances can be unreliable.
>
>    The dataset includes a variety of instances, from toy examples to real-world and industry problems (please see the **Novelty** section of our global response for more details).
>
> ----
> **Questions**
>
> 1. Reviewer’s question: Correctness of mathematical formulation. Are error metrics reliable?
>
>    Our evaluation of OptiMUS ensures that only correct solutions are counted as solved by ensuring a proposed solution passes human-supervised tests and matches the ground truth optimal value. In practice, we imagine domain experts using OptiMUS would use LLM-generated testing scripts as a starting point to write supervised tests, saving time compared to writing tests (or solver code) directly.
>
> 2. Reviewer’s question: How does OptiMUS set internal solver parameters? Is the output of OptiMUS stable?
>
>    Indeed, modern optimization solvers do have many parameters to set that control heuristics that are used internally and can affect the speed of finding the solution. However, one of the glories of standard optimization solvers is that they are guaranteed to find a solution if one exists. OptiMUS uses default parameters. Setting these parameters better to minimize solve time is an interesting problem that has a large literature. We have discussed the stability of OptiMUS in response to question 7.
>
> 3. Reviewer’s question: Does OptiMUS check optimality of the output?
>
>    In our evaluations, we do check for optimality of the output (Please see answer to question 1). However, in practice, it is not possible to check for optimality until we have solved the problem.
>
> 4. Reviewer’s question: The instances in NLP4LP are from textbooks. Chance of being in the training data; Instances can be very different from real problems and they have relatively small scale.
>
>    It is possible that the textbooks appeared in the training set of GPT4. However,
>
>    a) the SNOPs do not appear in the training dataset, as they were developed more recently (by the present authors)
>
>    b) Code solutions to these problems do not exist in the textbooks (or, to our knowledge, on the internet). Moreover, as discussed in the **Novelty** section of our global response, our dataset does include real-world problems.
>
> 5. Reviewer’s question: Why are success rates higher than execution rates?
>
>    Apologies for the confusion here. We swapped two legends of Figure 5 by mistake. We have corrected the mistake in our revision.
>
> 6. Reviewer’s question: missing references and Latex issues
>
>    Thank you for pointing these out. We have fixed them in the revised version
>
> 7. Reviewer’s question: Can the authors comment on reproducibility?
>
>    In our experiments, we use the "temperature" parameter of LLM to control the stability of output. Setting the temperature to 0 improves stability but reduces performance, as OptiMUS uses the diversity of augmentations to improve success rate. However, since we are calling APIs, the output in general will be different when the underlying LLM changes. We do not find this source of variability to have had a significant impact during the period since submission of this paper.
>
> Thanks again for your valuable feedback. All the concerns posed by the reviewer have been thoroughly addressed. If the reviewer find our subsequent response and revisions satisfactory, we kindly ask for an updated evaluation of our paper.

---

### Official Review · Reviewer_QHcC · 2023-10-31

**Soundness:** 2 fair
**Presentation:** 3 good
**Contribution:** 3 good
**Rating:** 5
**Confidence:** 3

**Summary:**

The authors tackled a challenging task of helping the modeling part of optimization problems via LLMs: they try to collect data, give some solid examples and prompting concepts, and show the performance of using the concept.

**Strengths:**

- Well-motivated problem and explanations of the paper content.
- The dataset collection for the challenging task with the SNOP (structured natural-language optimization problem), with experimental evaluations of several aspects (GPT-3.5 and GPT-4, some ablation).

**Weaknesses:**

- Although the LLM-based framework has good performance, the contributions in this paper seem to be experimental findings (rather than some new methods or theoretical analytics).

**Questions:**

- Please clarify or comment on the two metrics: success rate and execution rate. At first glance, the success rate seems to include execution rates (i.e., successes are only achievable when executable). Is this correct? In addition, in Fig. 5 of CPT4 + Prompt + Debug + Supervised Test, the execution rate coincides with the success rate. This bar is completely different from others. So, the authors are better to give some explanations (or intuitions).
- Some basic questions in the pipeline to follow the concept of OptiMUS:
    - In each part involving LLMs (e.g., the formulation in markdown, code generations, test generations), do LLMs (i.e., GPT-3.5, GPT-4) always succeed? Give some errors in practice.  Of course, I can believe that they `can` do them, but I’m interested in how we can believe their outputs and how often we should take care of them in the pipeline.
- Minor comments
    - I’m not exactly sure the reason, but some LaTeX links are not correctly inserted (some points are ??, maybe related to the appendix link). They should be fixed for readability.

---

> ### Author Response · Authors · 2023-11-18
> **Response to reviewer QHcC**
>
> **Response to reviewer QHcC**
>
> We thank the reviewer for the constructive suggestions.
>
> ----
> **Weaknesses**
>
> 1. Reviewer’s concern: The contributions in this paper seem to be experimental findings (rather than some new methods or theoretical analytics)
>
> We respectfully refer the reviewer to the **Novelty** section in our global response.
>
> ----
> **Questions**
>
> 1. Reviewer’s question: Success rate and execution rate labels are confusing
>
>    We are sorry for the confusion here. The legend labels were swapped by mistake. We have fixed this error in the revised version and we kindly refer the reviewer to the **Clarification** section in our global response.
>
> 2. Reviewer’s question: Do LLMs always succeed in different parts of the pipeline?
>
>    We appreciate the great question. Here is a more detailed list of how and where failures happen in our tests:
>
>    -  Modeling: 3 instances where the optimization model was missing constraints or had incorrect constraints.
>    - Unit mismatch: 1 instance where the output’s unit (cost per click) was different from the desired unit (cost per 1000 clicks). This could be resolved by clarifying the unit in the objective.
>    - Code omission: 1 instance where the model simply replaced some parts of the code with “...” during revision, resulting in an incorrect output
>    -  Debugging failure: 3 instances where the model was not able to debug the code and make it executable
>    -  Output format: 1 instance where the format of the generated output JSON did not match the desired format presented in the SNOP
>
> 3. Reviewer’s question: Latex link issues
>
>    Thank you for pointing this out. We fixed the links in the revision.
>
> Thanks again for reading the paper and providing us with valuable feedback. We have addressed all of the reviewer’s questions and listed our methodological and scientific contributions in our global response. If the reviewer is satisfied with our response and the revision, we respectfully request that the reviewer update the score of this paper.

---

> > ### Comment · Reviewer_QHcC · 2023-11-20
> > **Thank you for your responses**
> >
> > I appreciate the feedback from the authors.
> >
> > To be honest, they answered my questions and partially clarified my concerns (e.g., explanations of experiments). However, I will keep my score because I have some concerns about the novelty and contributions (i.e., a new paragraph written in blue and some new figures) after the revision. Particularly, NLP4LP and the improvements are interesting, but they seem to be particularly new results, so we should carefully review and discuss them in my personal view.

---

> > > ### Author Response · Authors · 2023-11-20
> > >
> > > Thank you for your prompt response. We would love to hear more about your other concerns and address them.

---

### Author Response · Authors · 2023-11-18
**Response to all the reviewers**

**Response to all the reviewers**

First, we sincerely appreciate all the reviewers for their efforts in reviewing our paper and their valuable suggestions. We find these suggestions very important and believe they will help further improve the quality of our paper.

We have noticed that most of the reviewers are concerned with **1)** the novelty of our proposed approach and **2)** our experiment setup and evaluation. We have updated our experiment setup in the paper and also added several clarifications **marked blue** in the revised version. Here is a summary of our novelty, and some clarifications on our experiments.

----
**Novelty**

We notice that several reviewers have expressed their concern about the novelty of our approach. We believe that developing an agent for optimization tasks, together with the prompt engineering and tuning needed to get it to work, is already a contribution to both the LLM and OR communities.

However, we believe that our novelty is beyond this: we are presenting a *scalable* methodology for the use of LLMs in optimization.

1. Instance abstraction and data-independence: There have been several previous attempts to use LLMs [1,2,3,4,5] to solve real-life optimization or math problems. Although these attempts provide novel ideas and valuable insights, they all have the same failing, which limits their use to toy examples of optimization problems: the input to the LLM includes the problem data as well as the problem description. The success of OptiMUS, conversely, is *independent* of the size of the problem data. Our revision includes a new figure (Figure 2), that shows solvers can fail for large-scale problems if data is passed to the LLM directly.

2. NLP4LP dataset: The other contribution of our paper is the development of the NLP4LP dataset. The dataset currently contains 61 problem instances (please see our anonymous repo) and is still growing. Moreover, many of these instances are not just toy problems. Several of the instances from [6] that we include in NLP4LP are real industrial problems (with mock data). Of course, once OptiMUS solves a problem, it can solve that problem equally well for data of arbitrary size, and so it can solve the original industrial application. We believe that NLP4LP can act as an important fundamental dataset to accelerate research on optimization modeling from natural language representations.

3. Test-based optimization: Another contribution of OptiMUS is proposing a new approach to solve optimization problems by introducing supervised and automated tests. OptiMUS allows users to tell how they want the solution to be by creating supervised-tests, rather than directly defining a mathematical model.

----
**Experiments and Evaluation**

We make the following updates/clarifications. We also kindly refer the reviewer to the revised version of our paper.

Clarifications:

1. We fixed the legend of Figure 5 by swapping success rate and execution rate. In the original figure two legends are mistakenly swapped
2. We added more details on our experiment setup to the revised paper, including:
   1. We set the maximum number of iterations to 5.
   2. We use Gurobi for our solver type.
   3. We get the ground-truth solutions either from textbook solution manuals or by solving the problems ourselves.
   4. We evaluate the model by making sure it passes our supervised testing scripts.

More updates:

1. We have further augmented our NLP4LP dataset to 61 instances
2. We improved the prompts, and OptiMUS is now able to achieve a higher performance using the new prompts. The evaluation figures are updated accordingly. Moreover, using the new prompts allows OptiMUS to improve performance with increasing debugging iterations.
3. We added two more plots, one on the distribution of the length of the SNOPs, and the other one on the distribution of the number of tokens generated by OptiMUS.

We sincerely hope that our additional experiments can address the reviewers' concerns. Again we appreciate all the reviewers' efforts and valuable suggestions in the review process.

----
**References**

[1] Guo, P. F., Chen, Y. H., Tsai, Y. D., & Lin, S. D. (2023). Towards Optimizing with Large Language Models. *arXiv preprint arXiv:2310.05204*.

[2] Yang, C., Wang, X., Lu, Y., Liu, H., Le, Q. V., Zhou, D., & Chen, X. (2023). Large language models as optimizers. *arXiv preprint arXiv:2309.03409*.

[3] Liu, S., Chen, C., Qu, X., Tang, K., & Ong, Y. S. (2023). Large Language Models as Evolutionary Optimizers. *arXiv preprint arXiv:2310.19046*.

[4] He-Yueya, J., Poesia, G., Wang, R. E., & Goodman, N. D. (2023). Solving math word problems by combining language models with symbolic solvers. *arXiv preprint arXiv:2304.09102*.

[5] Zhang, C. E., Collins, K. M., Weller, A., & Tenenbaum, J. B. (2023). AI for Mathematics: A Cognitive Science Perspective. *arXiv preprint arXiv:2310.13021*.

[6] Williams, H. P. (2013). *Model building in mathematical programming*. John Wiley & Sons.

---

### Meta-Review · Area_Chair_AFPi · 2023-12-09

**Metareview:**

The paper describes a new framework based on LLMs to translate natural language descriptions of LPs and MILPs into code that is then fed to a solver to find a solution and possibly debug it.  This is an unusual paper in the sense that it does not propose new optimization or machine learning algorithms, but it describes a framework to leverage and integrate LLMs.  This is highly innovative work that tries to make optimization more accessible.  However the paper does not really describe the algorithms that make up the framework to translate natural text problem descriptions into code with verification and debugging.  It describes the framework at a very high level, and it reports good results with a newly contributed benchmark, but the algorithmic details are missing. At the end of the day, a publication should include a detailed description of the proposed algorithms.

**Justification For Why Not Higher Score:**

The algorithmic details of the proposed framework are missing.

**Justification For Why Not Lower Score:**

N/A

---

### Decision · Program_Chairs · 2024-01-16

Reject